# Changing trends in ophthalmological emergencies during the COVID-19 pandemic

**José Escribano Villafruela** [iD] *, **Antonio de Urquía Cobo, Fátima Martín Luengo** [☯], **Víctor Antón Modrego** [☯], **María Chamorro González-Cuevas** [☯]

Department of Ophthalmology, Hospital General Universitario Gregorio Marañón, Madrid, Spain

☯ These authors contributed equally to this work.
* josecarlos.escribano@salud.madrid.org

**Data Availability Statement:** All relevant data are within the manuscript and its Supporting Information files.

## Abstract

On March 11, 2020, the World Health Organization declared COVID-19—the infectious disease caused by SARS-CoV-2—a pandemic. Since then, the majority of countries—including Spain—have imposed strict restrictions in order to stop the spread of the virus and the collapse of the health systems. People's health care–seeking behavior has exhibited a change, not only in those months when the COVID-19 control measures were strictest, but also in the months that followed. We aimed to examine how the trends in ophthalmological emergencies changed during the COVID-19 pandemic in one of the largest tertiary referral hospitals in Spain. To this end, data from all the patients that attended the ophthalmological emergency department during the pandemic period—March 2020 to February 2021—were retrospectively collected and compared with data from the previous year. Moreover, a comparison between April 2020—when the restrictions were most severe—and April 2019 was made. A total of 90,694 patients were included. As expected, there was a decrease in the number of consultations. There was also a decrease in the frequency of conjunctival pathology consultations. These changes may bring to light not only the use that people make of the emergency department, but also the new trends in ophthalmological conditions derived from the hygienic habits that the COVID-19 pandemic has established.

## Introduction

At the end of 2019, the Chinese government announced the exponential increase in SARS-CoV-2 infections in the city of Wuhan [1]. This virus was identified as the pathogen of a new disease named COVID-19, whose symptoms were harmless in some patients but could cause severe respiratory failure and even death in others [2]. This virus easily spread worldwide; in February 2020, the first cases in Spain appeared. On March 11, 2020, World Health Organization (WHO) officially declared COVID-19 a pandemic [3]. Three days later, the Spanish government declared the country in a "state of alarm," canceling all commercial and financial activities considered non-essential and ordering the lockdown of residents at home [4]. Later, from March 28 to April 12, 2020, additional restrictions were included: all displacements for non-essential workers were banned, and all the medical and surgical assistance considered non-emergency were canceled [5].

**Funding:** The authors received no specific funding for this work.

**Competing interests:** NO authors have competing interests.

Madrid was the epicenter of the outbreak in Spain, with the worst records of COVID-19 infections in the country. Hospital General Universitario Gregorio Marañón is one of the biggest tertiary hospitals in Spain and it provides medical attention to a healthcare area greater than 300.000 people. In addition, the Ophthalmological Emergency Department of the hospital embraces two more healthcare areas to give medical attention to a total population of 700.000 people in Madrid. The hospital attended to a large portion of patients with COVID-19 during the pandemic, so most medical staff members were rerouted to deal with this new situation. By contrast, the Ophthalmology Department kept attending all the ocular emergencies and non-deferrable surgical procedures such as retinal detachments, ocular perforations, or acute glaucoma surgery. The state of alarm ended on June 21, 2020, after a period of 98 days. Strict lockdown lasted for 42 days (March 14–April 26, 2020). We aimed to examine the changes in ophthalmological emergencies during the outbreak of the COVID-19 pandemic as well as in the months after the lockdown had finished. We hypothesized that the lockdown altered the trends of ocular pathology consultations, and the change in hygienic habits—wearing masks and hand washing—contributed to this modification.

## Materials and methods

In this retrospective, single-center, observational study, we analyzed the records of all the patients referred to the ophthalmological emergency department of Hospital General Universitario Gregorio Marañón (Madrid, Spain) from January 2015 to March 2021. Data was provided by the Admissions and Clinical Documentation Department of our hospital in an anonymized format to avoid a privacy data breach. The number of eye emergency visits per month since 2015, the diagnoses issued with automated International Statistical Classification of Diseases and Related Health Problems, Tenth Revision (ICD-10) codes since September 2019, and the number of COVID-19-positive cases in our hospital since the outbreak of the pandemic were included in the records.

The study was carried out in accordance with the Declaration of Helsinki (1989) and it was approved by the Ethical Committee of our hospital (CPMP/ICH/135/95). Informed consent form was waived in accordance with the Ethical Committee because of the anonymity of the data and the retrospective character of the research.

In this study, the conditions diagnosed were categorized for comparison into 11 groups: conjunctival, corneal, lens, lids and orbit, lacrimal system, ocular inflammation, vitreous, retina and choroid, neuro-ophthalmology, and others.

The number of visits per month from March 2020 to February 2021, named "pandemic year," were compared with the median number of visits per month from January 2015 to February 2020. In addition, the relation between the number of visits per month of the pandemic year and the number of patients with a positive COVID test from our hospital each month of this year was analyzed. We studied the conditions and groups reported in April 2020, as a representative month of the lockdown, and compared them with the ones of April 2019. Furthermore, we selected a 6-month period during which time the restrictions had lessened (September 2020–February 2021) as representative of the months after the outbreak and compared it with an equal period of time of the non-pandemic period (September 2019–February 2020).

Statistical analysis was performed by using SPSS software version 25 for Windows (IBM Corporation, Armonk, NY, USA). Descriptive statistics were used to summarize the mean values and standard deviations of all numerical data. The $\chi^2$ test was used to compare frequencies of categorical variables. $P < 0.05$ was considered significant. Regression analysis was performed by using the coefficient of determination ($R^2$) to assess the proportion of the variation in the dependent variable that is predictable from the independent variable.

**Table 1. Number of visits to the ophthalmological emergency department since 2015.**

| YEAR | JAN | FEB | MAR | APR | MAY | JUN | JUL | AUG | SEP | OCT | NOV | DEC |
|------|-----|-----|-----|-----|-----|-----|-----|-----|-----|-----|-----|-----|
| 2015 | 1136 | 1119 | 1287 | 1232 | 1434 | 1458 | 1203 | 1026 | 1193 | 1228 | 1167 | 1167 |
| 2016 | 1130 | 1175 | 1225 | 1337 | 1369 | 1356 | 1184 | 1006 | 992 | 1050 | 1082 | 1081 |
| 2017 | 946 | 863 | 1112 | 928 | 1571 | 1512 | 1442 | 1284 | 1364 | 1390 | 1343 | 1431 |
| 2018 | 1292 | 1274 | 1423 | 1417 | 1469 | 1440 | 1452 | 1288 | 1378 | 1472 | 1446 | 1498 |
| 2019 | 1496 | 1449 | 1681 | 1480 | 1577 | 1527 | 1536 | 1354 | 1414 | 1463 | 1393 | 1372 |
| 2020 | 1476 | 1381 | 501 | 302 | 807 | 1030 | 989 | 889 | 761 | 801 | 838 | 988 |
| **2021** | 674 | 843 | | | | | | | | | | |

## Results

The total number of visits during the pandemic year (March 2020–February 2021) was 9,423, which is markedly less compared with the previous year (Table 1). The month with the greatest difference was April 2020, with 1,178 fewer visits compared with the year before. In June 2020, there was a knock-on effect, with 1,030 visits; this month had the most consultations in the pandemic year. The number of visits over the next months remained stable (674–988), maintaining the trend of fewer visits compared with the previous years. There were significant differences in the number of visits each month during the pandemic year compared with the mean number of consultations per month between January 2015 and February 2020 ($p < 0.05$) (Fig 1).

Linear regression analysis was performed to assess the correlation between the number of COVID-19-positive cases diagnosed each month in our hospital and the number of

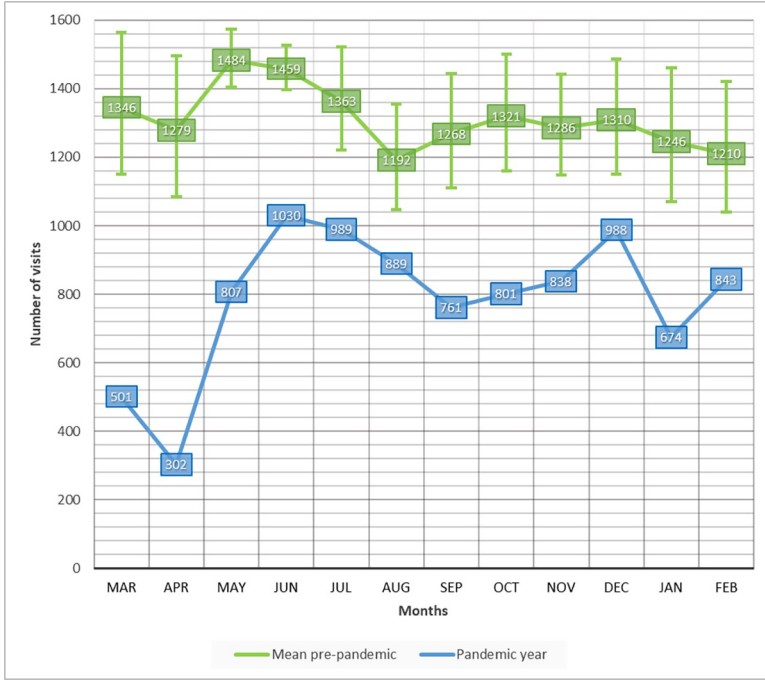

**Fig 1. The number per visits per month in the ophthalmological emergency department.** The mean number of visits per month from January 2015 to February 2020 is shown in green. The number of visits per month from March 2020 to February 2021 is shown in blue. Error bars represented the maximum and minimum number of visits each month from January 2015 to February 2020.

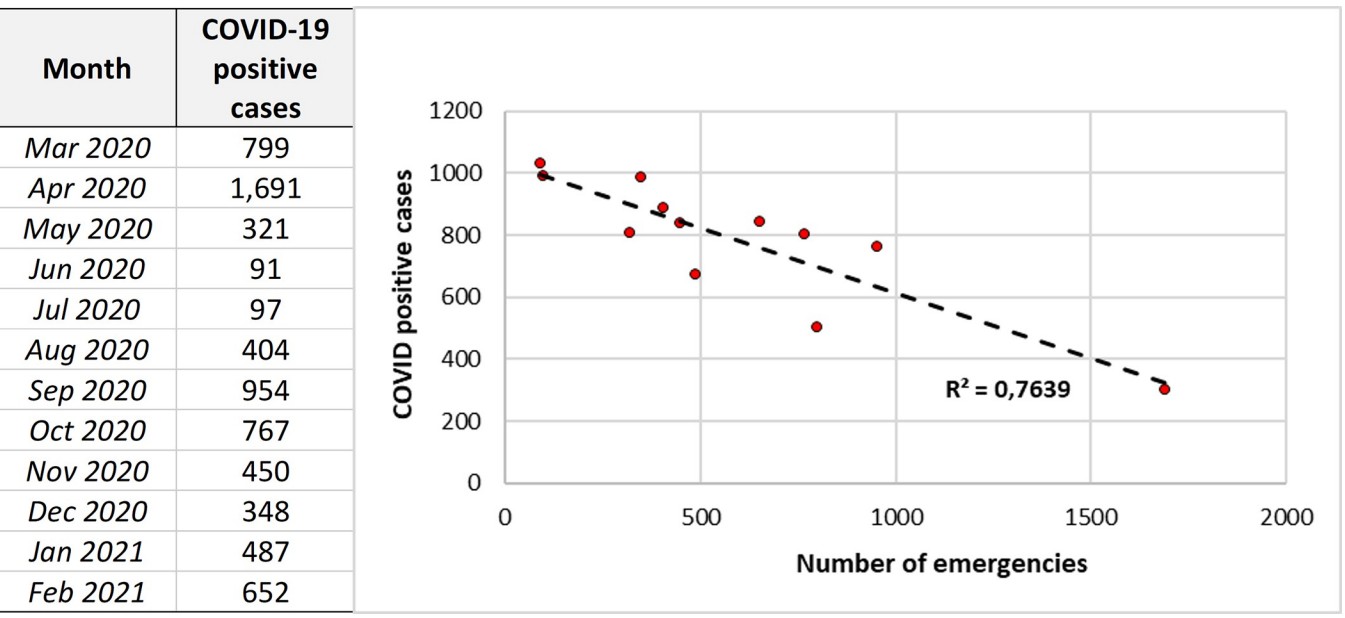

| Month | COVID-19 positive cases |
|-------|-------------------------|
| Mar 2020 | 799 |
| Apr 2020 | 1,691 |
| May 2020 | 321 |
| Jun 2020 | 91 |
| Jul 2020 | 97 |
| Aug 2020 | 404 |
| Sep 2020 | 954 |
| Oct 2020 | 767 |
| Nov 2020 | 450 |
| Dec 2020 | 348 |
| Jan 2021 | 487 |
| Feb 2021 | 652 |

**Fig 2. Regression analysis of the number of COVID-19-positive cases and the number of visits per month during the pandemic year.**

ophthalmological emergencies attended during the pandemic year. The number of ocular emergency consultations decreased in the months when there were more COVID-19-positive cases ($R^2 = 0.76$) (Fig 2). April 2020, the month with the fewest consultations in our database (302), had the highest number of COVID-19-positive tests (1,691).

During the most representative month of the lockdown (April 2020), ocular emergencies decreased drastically from 1,481 in 2019 to 302. Analyzing the different groups of conditions diagnosed, we found a statistically significant decrease in consultations for conjunctival diseases (415 [28.0%] to 54 [17.9%]; $p < 0.001$). By contrast, the consultations for vitreous (111 [7.5%] to 35 [11.6%]; $p < 0.01$) or retina and choroid (51 [3.4%] to 22 [7.4%]; $p < 0.002$) pathologies increased significantly. Although no significant differences were found, we detected an increase in the number of emergencies for ocular inflammation (57 [3.8%] to 19 [6.3%]; $p = 0.05$) (Table 2).

The three most frequent diagnoses of April 2020 were keratitis, conjunctivitis, and a tie for posterior vitreous detachment and corneal ulcer. By contrast, the three most frequent diagnoses of the previous year (April 2019) were conjunctivitis, keratitis, and corneal ulcer. The percentage of emergencies corresponding to keratitis did not change from 2019 to 2020. It is worth noting that the percentage of uveitis rose to the fifth place (5.6%) in 2020 (Table 3).

When comparing the six months after the outbreak of the COVID-19 pandemic (September 2020–February 2021) with the same months of the year before, the number of visits almost halved, from 8,497 to 4,902. While there was a significant decrease in the conjunctival group (1,967 [23.1%] to 822 [16.8%]; $p < 0.001$), other groups, such as corneal pathology (2,716 [32.0%] to 1,824 [37.2%]; $p < 0.001$), ocular inflammation (349 [4.1%] to 267 [5.4%]; $p < 0.001$), vitreous pathology (541 [6.4%] to 388 [7.9%]; $p < 0.01$), and neuro-ophthalmology (174 [2.0%] to 133 [2.7%]; $p < 0.02$) increased significantly (Table 4).

During this pandemic period, as we noticed in April 2020, corneal pathology was still the top group, and the three most frequent diagnoses were corneal ulcer, keratitis, and conjunctivitis. By contrast, the most frequent diagnoses in the non-pandemic period studied were conjunctivitis, followed by keratitis and corneal ulcer. Hordeola were more infrequent during the

**Table 2. Number of diagnoses categorized by group in the representative month of the pandemic outbreak (April 2020) compared with the same month of the year before.**

| Groups | April 2019 | April 2020 | p |
|---|---|---|---|
| Conjunctiva | 415 (28.0) | 54 (17.9) | < **0.001** |
| Cornea | 495 (33.4) | 101 (33.4) | 0.995 |
| Lens | 38 (2.6) | 5 (1.7) | 0.347 |
| Glaucoma | 10 (0.7) | 5 (1.7) | 0.089 |
| Orbit and eyelids | 162 (10.9) | 32 (10.6) | 0.862 |
| Lacrimal system | 16 (1.1) | 2 (0.7) | 0.508 |
| Ocular inflammation | 57 (3.8) | 19 (6.3) | 0.055 |
| Vitreous | 111 (7.5) | 35 (11.6) | ***0.018*** |
| Retina and choroid | 51 (3.4) | 22 (7.4) | ***0.002*** |
| Trauma | 48 (3.2) | 13 (4.3) | 0.354 |
| Neuro-ophthalmology | 24 (1.6) | 8 (2.6) | 0.220 |
| Others | 54 (3.6) | 6 (2.0) | 0.145 |
| Total | 1,481 | 302 | |

The visits are presented as the number (percentage).

The $\chi^2$ test was used to compare frequencies and $p < 0.05$ were considered significant.

pandemic period (4.0%) compared with the non-pandemic period (5.0%). On the other hand, the percentage of uveitis rose from 2.9% in the non-pandemic period to 4.0% in the pandemic period (Table 5).

## Discussion

The decrease in ophthalmological emergency visits during the COVID-19 pandemic compared with the 6 years before is remarkable, particularly in April 2020, because the restrictions were more severe than the rest of the period. During the SARS epidemic in 2003 in Taiwan, an analogous situation was described. As researchers suggested, the fear of contracting the disease would discourage the general population from going to the hospital [6]. The tendency to avoid attending to medical departments during the pandemic has already been reported in other studies, not only in ophthalmology services [7–9], but in units responsible for the management of more severe pathologies [10].

**Table 3. Leading diagnoses and their International Statistical Classification of Diseases and Related Health Problems, Tenth Revision (ICD-10) codes in the representative month of the COVID-19 pandemic outbreak (April 2020) compared with the same month of the year before.**

| Diagnosis April 2019 | n | Diagnosis April 2020 | n |
|---|---|---|---|
| Conjunctivitis (H10.3) | 325 (21.9) | Keratitis (H16.9) | 48 (15.9) |
| Keratitis (H16.9) | 235 (15.9) | Conjunctivitis (H10.3) | 35 (11.6) |
| Corneal ulcer (H16.0) | 123 (8.3) | Posterior vitreous detachment (H43.81) | 30 (9.9) |
| Corneal foreign body (T15.0) | 102 (6.9) | Corneal ulcer (H16.0) | 30 (9.9) |
| Posterior vitreous detachment (H43.81) | 99 (6.7) | Uveitis (H20) | 17 (5.6) |
| Hyposphagma (H 11.3) | 74 (5.0) | Hordeolum (H00.02) | 15 (5.0) |
| Blepharitis (H01.00) | 67 (4.5) | Corneal foreign body (T15.0) | 12 (4.0) |
| Hordeolum (H00.02) | 44 (3.0) | Hyposphagma (H 11.3) | 12 (4.0) |
| Ocular trauma (S05.9) | 41 (2.8) | Blepharitis (H01.00) | 6 (2.0) |
| No pathology (H53) | 37 (2.5) | Ocular trauma (S05.9) | 6 (1.7) |
| Uveitis (H20) | 36 (2.4) | Cutaneous herpes infection (B00.1) | 5 (1.7) |

**Table 4.  Number of diagnoses categorized by group in six representative months after the COVID-19 pandemic outbreak (September 2020–February 2021) compared with the same period of the year before.**

| Groups | Non pandemic | Pandemic | p |
|---|---|---|---|
| Conjunctiva | 1,967 (23.1) | 822 (16.8) | <**0.001** |
| Cornea | 2,716 (32.0) | 1,824 (37.2) | <**0.001** |
| Lens | 165 (1.9) | 120 (2.4) | 0.05 |
| Glaucoma | 74 (0.9) | 55 (1.1) | 0.123 |
| Orbit and eyelids | 1,102 (13.0) | 635 (13.0) | 0.980 |
| Lacrimal system | 89 (1.0) | 40 (0.8) | 0.186 |
| Ocular inflammation | 349 (4.1) | 267 (5.4) | <**0.001** |
| Vitreous | 541 (6.4) | 388 (7.9) | **0.001** |
| Retina and choroid | 412 (4.8) | 235 (4.8) | 0.887 |
| Trauma | 486 (5.7) | 268 (5.5) | 0.541 |
| Neuro-ophthalmology | 174 (2.0) | 133 (2.7) | **0.013** |
| Others | 422 (5.0) | 114 (2.3) | <**0.001** |
| Total | 8,497 | 4,902 | |

The visits are presented as the number (percentage). The $\chi^2$ test was used to compare frequencies and $p < 0.05$ was considered significant.

Our data showed that conjunctivitis was the most common diagnosis in the ophthalmological emergency department before the pandemic; this finding is consistent with other studies [11,12]. As other authors have remarked [8,10,13], conjunctival pathology consultations decreased significantly during the COVID-19 pandemic. In April 2020, conjunctival pathology represented 17.9% of ophthalmology consultations in our hospital, compared with 28% in April 2019. This trend seemed to continue during the entire pandemic period: conjunctivitis was the most common ophthalmology consultation in the non-pandemic period, while it dropped to the third position during the pandemic. This changing trend could be explained by multiple factors.

On the one hand, some authors have reported a decline in the number of "minor emergency visits" such as conjunctivitis [8], suggesting that the patient's fear of contagion could compromise health care seeking during the pandemic. These minor emergency visits could be managed by the patients using other services such as private clinics or via telemedicine with their general practitioner due to the reluctance to consult in the hospital. However, we could

**Table 5.  Leading diagnoses and their International Statistical Classification of Diseases and Related Health Problems, Tenth Revision (ICD-10) codes in six representative months after the COVID-19 pandemic outbreak (September 2020–February 2021) compared with the same period of the year before.**

| Diagnosis non pandemic | n | Diagnosis pandemic | n |
|---|---|---|---|
| Conjunctivitis (H10.3) | 1,438 (16.9) | Corneal ulcer (H16.0) | 650 (13.3) |
| Keratitis (H16.9) | 1,139 (13.4) | Keratitis (H16.9) | 633 (12.9) |
| Corneal ulcer (H16.0) | 751 (8.8) | Conjunctivitis (H10.3) | 515 (10.5) |
| Corneal foreign body (T15.0) | 517 (6.1) | Corneal foreign body (T15.0) | 387 (7.9) |
| Posterior vitreous detachment (H43.81) | 482 (5.7) | Posterior vitreous detachment (H43.81) | 354 (7.2) |
| Hyposphagma (H 11.3) | 426 (5.0) | Hyposphagma (H 11.3) | 224 (4.6) |
| Hordeolum (H00.02) | 367 (5.0) | Blepharitis (H01.00) | 209 (4.3) |
| Ocular trauma (S05.9) | 351 (4.1) | Uveitis (H20) | 195 (4.0) |
| Blepharitis (H01.00) | 340 (4.0) | Hordeolum (H00.02) | 194 (4.0) |
| Uveitis (H20) | 244 (2.9) | Ocular trauma (S05.9) | 191 (3.9) |
| No pathology (H53) | 239 (2.8) | Migraine (G43.9) | 88 (1.8) |

not find any publication or registration regarding an increase in consultations of ocular emergencies with real data using these features in our healthcare area.

Furthermore, hand washing has been demonstrated to be an effective measure to prevent transmission of viruses and other pathogens [14]. Washing hands and social distancing, both the main preventive actions of COVID-19 and recommended since the beginning of the pandemic [15,16], are also important means to stop conjunctivitis contagion, because viral conjunctivitis—the main form of conjunctivitis—is highly contagious and can be prevented in this way [17,18]. During the COVID-19 pandemic, adults have been more conscious about the significance of washing hands and this practice has gained importance [19], which could have diminished the incidence of this pathology in the emergency department.

On the other hand, COVID-19 is also a conjunctivitis etiological agent. The percentage of ocular manifestations in patients infected by SARS-CoV2 has been reported at approximately 11%, with ocular pain, redness, and follicular conjunctivitis as the main ophthalmic features found [20]. Nevertheless, due to the minor emergency nature of conjunctivitis and the fear of consultation, as previously discussed, the frequency of the disease in our data did not increase during the pandemic period.

Since face masks became mandatory, many have suggested that they could cause eye problems such as a hordeolum [21]. These authors argue that masks accelerate tear evaporation and increase the symptoms of dry eye, as other studies have reported [22–24]. This dry eye is related to blepharitis and obstruction of the meibomian glands, a phenomenon that eventually causes a hordeolum [25,26]. We have analyzed the months since mask use became mandatory on July 26, 2020, so its use was not so widespread in the population before that date. Our data showed that there was not a significant percentage increase in hordeola since masks became mandatory (194 [4.0%]) compared with the non-pandemic period studied (367 [5.0%]). This could be explained by an underdiagnosis of minor pathologies during those months. Moreover, hordeola can be managed by the general practitioner, so not many reach the ophthalmological emergency department.

Our study showed that there was no percentage increase in ocular trauma during the pandemic period. The number of cases of ocular trauma was 268 (5.5%) during the pandemic period, whereas this number was 486 (5.7%) during the non-pandemic period. This change could be explained by a decrease in traffic accidents [27], assaults due to fights [28,29], and workplace accidents [30–32]. On the other hand, there was an increase in domestic activities by unqualified personnel that counteracted this drop in injuries [33–35] and it is likely attributed to centralization of ophthalmic services during the pandemic crisis in ophthalmic services during the COVID-19 pandemic [36].

Different studies have described thrombotic complications of COVID-19 like pulmonary embolism, disseminated intravascular coagulation, stroke, and digit and limb infarcts [37]. There have also been case reports about ocular central vein occlusion in association with this new condition [38]. There is increasing evidence suggesting possible retinal microvascular sequelae in patients infected by SARS-CoV-2, assuming retinal microvasculopathy develops in 10% of infected people [39]. Contrary to our expectation, we did not find differences in ocular vein occlusions (24 cases [0.3%] vs. 17 cases [0.3%]) or arterial occlusions (10 cases [0.1%] vs. 2 cases [0%]) between the non-pandemic and the pandemic periods in the patients who attended the ophthalmologic emergency department. Other studies support our findings of no differences in retinal vascular occlusion during the lockdown [8,20].

We observed a percentage increase in the number of more relevant diagnoses such as vitreous pathology, retinal pathology, and ocular inflammation during April 2020. We found it interesting considering that this month was the one with the highest report of COVID-19-positive cases, emphasizing that the cases represented real ophthalmological emergencies. These

data may indicate the misuse of the ophthalmological emergency service before the COVID-19 pandemic [8,12]. Despite that, we did not find these differences to be clinically significant. Some authors have observed this same trend of avoiding emergency departments for symptoms that can be managed by the patients themselves or through other levels of assistance [40,41]. This may be related to the knock-on effect observed in June 2020, increasing the number of visits to 1,030, after the reduction because of the containment measures.

Our study was limited by the possibility of underdiagnosis of the most common ocular emergencies because of the use of different medical assistance as consultation in private practices or via telemedicine with their general practitioner, because of the reluctance of the population to consult in the hospital for fear of contagion. In addition, a single-center study is not the most suitable option to make these conclusions, but we have also considered that the big area of population we attend truly weighs upon the description of the changes in ocular emergencies during the pandemic period. The COVID-19 pandemic and statewide stay-at-home orders have created unprecedented changes to the health care system. Further investigation will be needed to assess if this trend continues.

## Conclusions

In conclusion, there was a significant reduction in our ocular emergency department consultations during the outbreak of the COVID-19 pandemic and the months thereafter. We found a significant percentage decrease in conjunctivitis that may be influenced by the use of face masks, hand washing, social distancing, and stay-at-home policies.

## Supporting information

**S1 Appendix.**
(DOCX)

**S1 Table.**
(PDF)

## Acknowledgments

We thank José María Bellón from the department of statistics and Daniel Toledano from the Admissions and Clinical Documentation Department of Hospital General Universitario Gregorio Marañón.

## Author Contributions

**Conceptualization:** José Escribano Villafruela, Fátima Martín Luengo, Víctor Antón Modrego, María Chamorro González-Cuevas.

**Data curation:** José Escribano Villafruela, Antonio de Urquía Cobo, Fátima Martín Luengo, Víctor Antón Modrego, María Chamorro González-Cuevas.

**Formal analysis:** José Escribano Villafruela.

**Funding acquisition:** José Escribano Villafruela.

**Investigation:** José Escribano Villafruela.

**Methodology:** José Escribano Villafruela, Antonio de Urquía Cobo.

**Project administration:** José Escribano Villafruela.

**Resources:** José Escribano Villafruela.

**Supervision:** José Escribano Villafruela, Antonio de Urquía Cobo.

**Validation:** José Escribano Villafruela, Antonio de Urquía Cobo.

**Visualization:** José Escribano Villafruela, Antonio de Urquía Cobo, Fátima Martín Luengo.

**Writing – original draft:** José Escribano Villafruela, Antonio de Urquía Cobo, Fátima Martín Luengo, Víctor Antón Modrego, María Chamorro González-Cuevas.

**Writing – review & editing:** José Escribano Villafruela.

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
