## [Decision Letter · Decision Letter 0]

11 Feb 2022

PONE-D-21-36763Changing trends in ophthalmological emergencies during the COVID-19 pandemicPLOS ONE

Dear Dr. Escribano Villafruela,

Thank you for submitting your manuscript to PLOS ONE. After careful consideration, we feel that it has merit but does not fully meet PLOS ONE’s publication criteria as it currently stands. Therefore, we invite you to submit a revised version of the manuscript that addresses the points raised during the review process.

Please read over the reviewer's comments carefully and make the required modifications. Take attention that the single-center study is not suitable to draw the manuscript's conclusions. Patients may have been referred to another hospital. There are no data or details on how local ophthalmology patient management was manage or whether these patients were able to be managed in another ophthalmology center. The data presented in this respect cannot therefore be evaluated on its own.

We look forward to receiving your revised manuscript.

Kind regards,

Adrienne Csutak, MD, PhD, MSc

Academic Editor

PLOS ONE

Journal Requirements:

“Unfunded studies”

Reviewers' comments:

Reviewer's Responses to Questions

**Comments to the Author**

1. Is the manuscript technically sound, and do the data support the conclusions?

Reviewer #1: No

Reviewer #2: Yes

2. Has the statistical analysis been performed appropriately and rigorously? 

Reviewer #1: Yes

Reviewer #2: Yes

3. Have the authors made all data underlying the findings in their manuscript fully available?

Reviewer #1: Yes

Reviewer #2: Yes

4. Is the manuscript presented in an intelligible fashion and written in standard English?

Reviewer #1: Yes

Reviewer #2: Yes

5. Review Comments to the Author

Reviewer #1: The authors aimed to examine how the trends in ophthalmological emergency cases changed during the COVID-19 pandemic in one of the largest tertiary referral hospitals in Spain. Data from more than 90 000 patients were collected retrospectively and compared with data from the previous year.

The conclusion is that the proportion of more severe ophthalmological conditions increased during the pandemic, while the number of cases decreased dramatically. In addition, the number of consultations for milder (e.g. conjunctival) diseases decreased partly for hygienical reasons and partly to fear of going to hospital.

The number of visits to ocular emergency consultations has fallen dramatically. The single-center study is not suitable to draw this conclusion. Patients may have been referred to another hospital. There are no data or details on how local ophthalmology patient management was manage or whether these patients were able to be managed in another ophthalmology center. The data presented in this respect cannot therefore be evaluated on its own.

Table 2 shows that the number of consultations for pathologies of the vitreous, retina and choroid has decreased to less than half, while the rate has increased by a statistically significant but only a few percent. Behind all statistically significant p-values is a difference of up to 6% between groups, which is not clinically significant in any way. This should be emphasized and all conclusions should be reconsidered and reassessed.

The message with the data in Table 1 is not transparent in this form (perhaps a graph is needed?).

The only real that is really clinically significant is that the number of monthly visits to the ophthalmology emergency department is decreased under COVID, which is an understandable fact; and is partly due to the population’s reluctance to consult the hospital for fear of contagion.

In summary, the conclusions of the study as presented are overstated and need to be reassessed and rewritten accordingly.

Reviewer #2: The topic of the article is current, analyzing the changes of ophthalmic care due to the COVID pandemic.

The topic of the article makes no mention of an important factor. Did patients have any opportunity for non-personal ophthalmic consultation/telemedicine during the pandemic? The number of patients appearing in the ohthalmic care system can be significantly affected by the possibility of a telephone consultation (one of the most common ophthalmic emergency conditions, conjunctivitis, can be treated during a pandemic by telephone and/or email consultation, followed by a prescription). This question should be added to the article.

The increase of uveitis cases during the pandemic period is mentioned in the last paragraph of the results chapter. What could have been the reason for this insignificant increase?

The number in the second column of the sixth row of Table 2 is incorrect (.7)

6. PLOS authors have the option to publish the peer review history of their article (what does this mean?). If published, this will include your full peer review and any attached files.

Reviewer #1: No

Reviewer #2: No

---

## [Author Response · Author response to Decision Letter 0]

26 Feb 2022

Ms. Ref. No.: PONE-D-21-36763

Title: Changing trends in ophthalmological emergencies during the COVID-19 pandemic

PLOS ONE 

Adrienne Csutak, MD, PhD, MSc

Academic Editor

PLOS ONE

Dear Professor Adrienne Csutak:

The authors of the manuscript would like to thank you and the reviewers for their constructive critique of our manuscript. The comments have helped us to improve the manuscript.

The responses to the questions are below. Furthermore, a new version of the cover letter, manuscript and the appendix have been uploaded with the changes highlighted in yellow.

Reviewer’s comments:

Reviewer #1: The authors aimed to examine how the trends in ophthalmological emergency cases changed during the COVID-19 pandemic in one of the largest tertiary referral hospitals in Spain. Data from more than 90 000 patients were collected retrospectively and compared with data from the previous year.

The conclusion is that the proportion of more severe ophthalmological conditions increased during the pandemic, while the number of cases decreased dramatically. In addition, the number of consultations for milder (e.g. conjunctival) diseases decreased partly for hygienical reasons and partly to fear of going to hospital.

The number of visits to ocular emergency consultations has fallen dramatically. The single-center study is not suitable to draw this conclusion. Patients may have been referred to another hospital. There are no data or details on how local ophthalmology patient management was manage or whether these patients were able to be managed in another ophthalmology center. The data presented in this respect cannot therefore be evaluated on its own.

Response:

Thank you for your comment. We agree with your statement: a single-center study is not the best method to truly confirm a decrease of emergency consultations in Madrid during the COVID pandemic. Some patients may have consulted in other centers, especially in the private practices, because of the fear of contagion at the hospital. Some other patients could have managed their ocular emergencies via telemedicine with their general practitioner. However, we could not find any publication or registration regarding an increase in consultations of ocular emergencies with real data using these features in our healthcare area.

Nevertheless, we would like to highlight that Hospital General Universitario Gregorio Marañon is one of the biggest public hospitals in Madrid and Spain. We provide medical attention to a big healthcare area, with a population of more than 300.000 people. The hospital also embraces two more healthcare areas to attend ophthalmological emergencies, including a population of 700.000 people in total. Because of this peculiarity, it could be said that our data provides information equivalent to that of three hospitals. Furthermore, during the first months of the pandemic, our hospital kept attending ocular emergencies by ophthalmologists in the Emergency service, in contrast with other public hospitals in our city, which remained closed. 

In conclusion, we consider your appreciation and we have added this limit to our study. A single-centre study is not the best design to make these conclusions, but we also consider that the big area of population we attend truly weighs upon the description of the changes in ocular emergencies during the pandemic period in our city.

Table 2 shows that the number of consultations for pathologies of the vitreous, retina and choroid has decreased to less than half, while the rate has increased by a statistically significant but only a few percent. Behind all statistically significant p-values is a difference of up to 6% between groups, which is not clinically significant in any way. This should be emphasized and all conclusions should be reconsidered and reassessed

Response: 

Thank you for your appreciation. As you suggest, we do not find this change clinically significant and modifications have been made in the manuscript to highlight this conclusion.

The message with the data in Table 1 is not transparent in this form (perhaps a graph is needed?).

Response:

Thank you for your suggestion. We have modified the style in Figure 1 and we have also added Table 1 as a graphical representation, in order to be more transparent. The mean number of visits per month from January 2015 to February 2020 is shown in green, with the maximum and minimum number of visits each month represented in error bars. The number of visits per month from March 2020 to February 2021 is shown in blue. 

The only real that is really clinically significant is that the number of monthly visits to the ophthalmology emergency department is decreased under COVID, which is an understandable fact; and is partly due to the population’s reluctance to consult the hospital for fear of contagion.

Response: 

Thank you for your appreciation. We agree with you that the main conclusion of the study is the decrease in the number of consultations. However, we consider that the reduction of conjunctivitis consultations was also clinically significant as other studies have assessed.

# Reviewer 2: The topic of the article is current, analyzing the changes of ophthalmic care due to the COVID pandemic.

The topic of the article makes no mention of an important factor. Did patients have any opportunity for non-personal ophthalmic consultation/telemedicine during the pandemic? The number of patients appearing in the ophthalmic care system can be significantly affected by the possibility of a telephone consultation (one of the most common ophthalmic emergency conditions, conjunctivitis, can be treated during a pandemic by telephone and/or email consultation, followed by a prescription). This question should be added to the article.

Response:

Thank you for your appreciation. We have added this question in the new manuscript as you suggest. In our hospital, telephone consultation was limited to the programmed medical activity of each ophthalmology subspecialty (retina, anterior segment, strabismus…). However, it was not implemented in the Ophthalmological Emergency Department, which remained always open. It is true that some patients could have solved their ocular emergencies via telephone with their general practitioner. However, we could not find any publication or registration about this matter in our healthcare area.

The increase of uveitis cases during the pandemic period is mentioned in the last paragraph of the results chapter. What could have been the reason for this insignificant increase?

Response:

Thank you for your comment. We do not think that the increase in uveitis diagnosis was clinically significant. This increase is expressed in relative but not in absolute terms. One of the reasons for this lower decrease is that patients with uveitis already knew the symptoms of their disease from previous episodes and they went to the Emergency Department to get the prescription of the drugs. 

The number in the second column of the sixth row of Table 2 is incorrect (.7)

Response: Thank you. As requested, we have fixed this mistake.

---

## [Decision Letter · Decision Letter 1]

12 May 2022

Changing trends in ophthalmological emergencies during the COVID-19 pandemic

PONE-D-21-36763R1

Dear Dr. Escribano Villafruela,

We’re pleased to inform you that your manuscript has been judged scientifically suitable for publication and will be formally accepted for publication once it meets all outstanding technical requirements.

Kind regards,

Adrienne Csutak, MD, PhD, MSc

Academic Editor

PLOS ONE

Additional Editor Comments (optional):

Reviewers' comments:

Reviewer's Responses to Questions

**Comments to the Author**

1. If the authors have adequately addressed your comments raised in a previous round of review and you feel that this manuscript is now acceptable for publication, you may indicate that here to bypass the “Comments to the Author” section, enter your conflict of interest statement in the “Confidential to Editor” section, and submit your "Accept" recommendation.

Reviewer #1: All comments have been addressed

Reviewer #2: (No Response)

2. Is the manuscript technically sound, and do the data support the conclusions?

Reviewer #1: Yes

Reviewer #2: Yes

3. Has the statistical analysis been performed appropriately and rigorously? 

Reviewer #1: Yes

Reviewer #2: Yes

4. Have the authors made all data underlying the findings in their manuscript fully available?

Reviewer #1: Yes

Reviewer #2: Yes

5. Is the manuscript presented in an intelligible fashion and written in standard English?

Reviewer #1: Yes

Reviewer #2: Yes

6. Review Comments to the Author

Reviewer #1: The revised manuscript is OK in its current form.

The revised manuscript is OK in its current form.

Reviewer #2: By reading the corrected manuscript, I accept the answers of the authors and recommend the article for publication.

7. PLOS authors have the option to publish the peer review history of their article (what does this mean?). If published, this will include your full peer review and any attached files.

Reviewer #1: No

Reviewer #2: No

---

## [Editor Report · Acceptance letter]

20 May 2022

PONE-D-21-36763R1 

Changing trends in ophthalmological emergencies during the COVID-19 pandemic 

Dear Dr. Escribano Villafruela:

I'm pleased to inform you that your manuscript has been deemed suitable for publication in PLOS ONE. Congratulations! Your manuscript is now with our production department. 

Kind regards, 

on behalf of

Dr. Adrienne Csutak 

Academic Editor

PLOS ONE